# In vitro characterisation of the MS2 RNA polymerase complex reveals host factors that modulate emesviral replicase activity

Alexander Wagner[1], Laura I. Weise[2] & Hannes Mutschler [1✉]

The RNA phage MS2 is one of the most important model organisms in molecular biology and virology. Despite its comprehensive characterisation, the composition of the RNA replication machinery remained obscure. Here, we characterised host proteins required to reconstitute the functional replicase in vitro. By combining a purified replicase sub-complex with elements of an in vitro translation system, we confirmed that the three host factors, EF-Ts, EF-Tu, and ribosomal protein S1, are part of the active replicase holocomplex. Furthermore, we found that the translation initiation factors IF1 and IF3 modulate replicase activity. While IF3 directly competes with the replicase for template binding, IF1 appears to act as an RNA chaperone that facilitates polymerase readthrough. Finally, we demonstrate in vitro formation of RNAs containing minimal motifs required for amplification. Our work sheds light on the MS2 replication machinery and provides a new promising platform for cell-free evolution.

[1] Department of Chemistry and Chemical Biology, TU Dortmund University, 44227 Dortmund, Germany. [2] Max Planck Institute of Biochemistry, 82152 Martinsried, Germany. ✉email: hannes.mutschler@tu-dortmund.de

*E*mesvirus zinderi, commonly called bacteriophage MS2, is one of the smallest viral pathogens known with its single-stranded 3.6 kilobase (kb) RNA genome encoding only four proteins[1,2]. Due to its simple architecture, MS2 finds extensive use as a model organism in various biological research fields and helped to reveal fundamental biological processes such as translational control, feedback inhibition and key aspects of viral life cycles[3–6]. Moreover, MS2 is used as a surrogate for human pathogenic RNA viruses to study the properties, stability and detectability of RNA viruses under different environmental conditions[7–9]. Apart from their relevance to RNA biochemistry and molecular biology, protein and RNA components of ssRNA coliphages such as Qβ and MS2 serve as attractive platforms for RNA imaging[10–12], RNA packaging and delivery[13–15] and molecular evolution and gene expression in cell-free systems[16–20].

Infection of male *Escherichia coli* cells by MS2 is dependent on the maturation protein (mp), which is present as a single copy in the MS2 capsid and enables binding of MS2 virions to the host's F-pili[6]. Following infection, the host translation machinery immediately begins synthesising the MS2 coat protein (cp), whose main function is to encapsulate newly synthesised viral (+) strands. Ribosomal readthrough events of this gene provide a brief opportunity for initiation of translation of the replicase (rep) gene, whose ribosome-binding site is otherwise sequestered in an "operator" hairpin RNA structure[21] and inhibited by long-range RNA-RNA interactions[22]. Similarly, timing and level of mp expression are controlled by the RNA folding kinetics of an untranslated leader sequence[23]. To initiate genome replication, the MS2rep subunit forms an active holocomplex, which likely consists of the *E. coli* elongation factors EF-Tu, EF-Ts, ribosomal protein S1 and a yet undefined host protein (Fig. 1)[1]. This complex synthesises new genomic (+) strand using antigenomic (−) strand intermediates as template. At later stages of the life cycle, cp dimers bind and stabilise the operator hairpin which ultimately leads to a complete suppression of translation of the rep gene[21]. Finally, expression of the lysis protein (lys) leads to the release of mature virions from the infected host, enabling new cycles of infection[6].

Emesviral replicases are amongst the most ancient RNA-dependent polymerase enzymes and share key characteristics with other viral RNA polymerases[24], and are therefore attractive model systems for drug discovery and mutational analyses. However, although MS2 is one of the main model systems for RNA viruses, fundamental aspects of its genome replication remain unclear to date. In particular, a host factor postulated almost 50 years ago[25,26], to be required for RNA replicase activity, remains unknown. Here, using our previously established reporter system for MS2 replicase activity in PURE systems[20], we are able to characterise several host factors that modulate the activity of the phage enzyme. Through stepwise reduction of PURE proteins, we identified initiation factor 1 (IF1) as a co-factor that stimulates the activity of the MS2 replicase core complex consisting of MS2rep, EF-Tu, EF-Ts and ribosomal protein S1. Furthermore, we found that initiation factor 3 (IF3) acts as an inhibitor for replicase activity, which has potential implications for the viral life cycle. Finally, we demonstrate that the MS2 replicase complex can evolve small replicable RNA species during the in vitro replication of the full-length genome. These new RNA scaffolds may be well-suited for the generation of robust self-replicating synthetic RNA systems in cell-free in vitro systems.

## Results

**Purified MS2 replicase is functional in recombinant in vitro translation systems.** To characterise the activity requirements of the MS2 replicase complex, we overexpressed the MS2rep subunit in *E. coli* and purified it by single-step immobilised metal ion affinity chromatography. In the one-step purification protocol, MS2rep co-eluted with the two potential host factors ribosomal protein S1 and the translation factor (TF) EF-Ts, both of which are components of the related Qβ replicase complex (Supplementary Fig. 1). Surprisingly, we noticed that EF-Tu, which is an essential and tightly binding host factor for the Qβ replicase complex that readily co-purifies with Qβrep, did not co-elute with the His-tagged MS2rep subunit. Thus, the assembly properties of the replicase core complexes between the genera *Qubevirus* (Qβ) and *Emesvirus* (MS2) show unexpected differences.

Next, we sought to explore if the MS2·S1·EF-Ts heterocomplex could be used to initiate transcription of a genuine MS2 template. To this end, we made use of our previously established MS2 RNA polymerase assay for the detection of MS2 replicase activity in recombinant in vitro transcription translation (PURE) systems[20,27]. In this assay, fluorescence from the fluorogenic ligand DFHBI-1T is observed upon (+) strand synthesis of the DFHBI-1T-binding broccoli aptamer[28] from its reverse complement, which has been embedded between the reverse complement of both untranslated regions of the MS2 genome ([F30-Bro(−)]$_{UTR(−)}$) (Fig. 2a). In agreement with our previous results for in situ expressed MS2rep[20], we observed a strong fluorogenic readout when both [F30-Bro(−)]$_{UTR(−)}$ and MS2·S1·EF-Ts were incubated in the commercially available PURExpress® system. We also observed aptamer synthesis, albeit at a lower level, in an

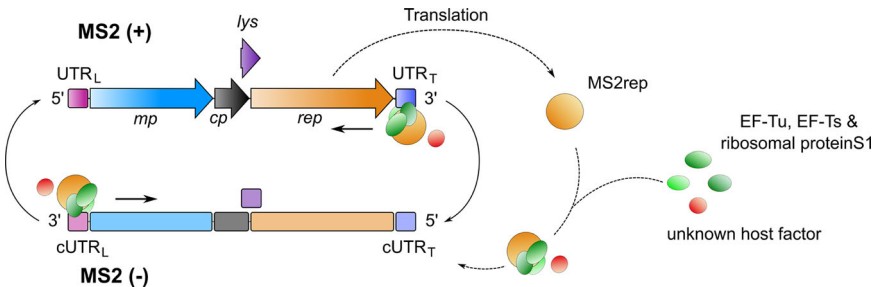

**Fig. 1 Replication cycle of MS2 RNA.** Scheme of replication for the ssRNA genome of MS2 bacteriophage. The 3569 nt genome encodes four proteins: the maturation protein (mp, blue) for host infection, the coat protein (cp, black) for genome encapsulation, the lysis protein (lys, purple) for cell lysis and the MS2 replicase subunit (rep, orange) for replication of the ssRNA genome. The four open reading frames on the plus strand (coloured arrows) are embedded between the leading untranslated region at the 5′ end (UTR$_L$, pink) and the trailing UTR at the 3′ end (UTR$_T$, dark blue). For the minus strand, the respective complementary sequences are depicted as rounded rectangles. Following translation of the MS2replicase subunit (MS2rep), formation of the holocomplex takes place. This complex is suspected to consist of host factors EF-Tu (green), EF-Ts (green), ribosomal protein S1 (green) and a putative additional host factor (red). RNA replication is initiated at the 3′ ends of the respective template strands, leading to the synthesis of the complementary, new template strand.

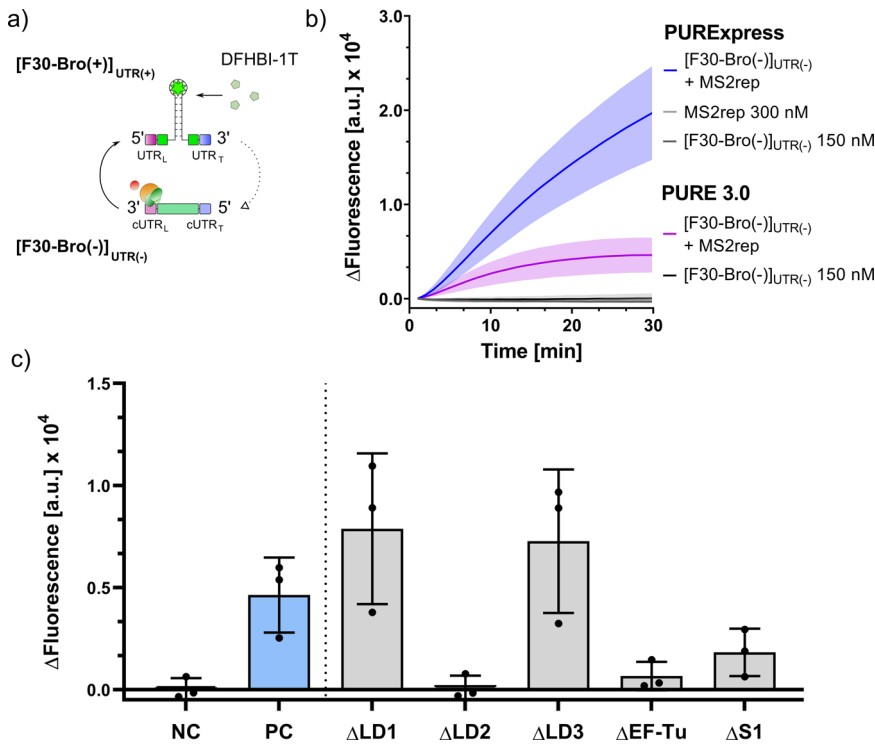

**Fig. 2 Effect of different PURE co-factors on [F30-Bro(+)]$_{UTR(+)}$ synthesis by MS2rep. a** The broccoli aptamer (F30-Bro) for fluorescence readout is incorporated into a replication scaffold based on the MS2 UTRs. Using [F30-Bro(−)]$_{UTR(−)}$ as template, the MS2 replicase complex synthesises [F30-Bro(+)]$_{UTR(+)}$ and the broccoli aptamer therein binds fluorogenic DFHBI-1T. **b** Replicase activity in commercially available PURExpress (blue), in an in-house PURE 3.0 (purple), as well as in negative controls either lacking MS2rep (black & dark grey) or the RNA template (light grey). Reactions were incubated at 37 °C for 30 min and fluorescence was measured every minute. **c** Endpoint fluorescence of depletion permutation assay for in-house PURE-based replication system, programmed with 50 nM [F30-Bro(−)]$_{UTR(−)}$ RNA. Reactions contained all components except MS2rep (NC, dark grey), all components (PC, blue), or all components except the indicated ones. Fluorescence was measured every minute over a 30-min time course at 37 °C. Error bars indicate standard deviation, based on three independent technical replicates.

in-house PURE system (PURE 3.0), which was prepared according to a previously established protocol[29] (Fig. 2b). Based on this result, we inferred that the PURE systems contained the host factors required to assemble the active MS2 holoenzymes.

To further narrow down the range of possible *E. coli* proteins that are required for MS2 replicase activity, we investigated the activity in the presence of different PURE protein fractions. In PURE 3.0, 30 of the 31 *E. coli* TFs are obtained after co-expression and purification of the TF genes from three large expression plasmids resulting in three protein fractions (LD1, LD2 and LD3)[29]. An additional enzyme mix contains 70S ribosomes as well as the elongation factor EF-Tu and the enzymes necessary to reconstitute an NTP regeneration system based on creatine phosphate (Supplementary Table 2). Initially, we tested MS2 replicase activity in reduced PURE 3.0 reactions (PUREred) based on the LD1-LD3 protein fractions. We also used a simplified enzyme mix containing only ribosomal protein S1 and EF-Tu because we did not expect ribosomes or kinase enzymes to be the missing co-factors based on the homology between MS2 and Qβ[30]. Both replicases amount to 30.5% sequence identity and 43.2% sequence similarity as determined by the EMBOSS Needle pairwise sequence alignment tool[31]. As anticipated, MS2 showed transcription of fluorogenic [F30-Bro(+)]$_{UTR(+)}$ in the PUREred environment (Fig. 2c). Next, we sought to narrow down the range of potential host factors by further omitting protein components from the PUREred setup. Hereby, we confirmed that TFs EF-Tu and the ribosomal protein S1 are critical co-factors required for full MS2 replicase activity, similarly to Qβ, as omission of both proteins would lead to a drastic loss in [F30-Bro(+)]$_{UTR(+)}$ synthesis (Fig. 2c). While

depletion of EF-Tu had a drastic effect on aptamer transcription, we still observed substantial transcription in the absence of added S1 protein, presumably due to the presence of S1 protein in the purified complex (Supplementary Fig. 1).

In addition to the expected dependency of the replicase on S1 and EF-Tu, we also observed an unforeseen impact on [F30-Bro(+)]$_{UTR(+)}$ synthesis upon omission of the individual LD protein fractions. Omission of both LD1 and LD3 had a modest enhancing effect on MS2rep activity which may not even be significant. This is likely caused by the altered crowding conditions in the absence of the protein factors, which may affect RNA folding and/or the formation of inert duplexes[32]. In contrast, omission of LD2 caused a drastic loss of [F30-Bro(+)]$_{UTR(+)}$ transcription activity (Fig. 2c) suggesting that at least one of its protein components enhances MS2rep activity. LD2 contains eight enzymes, four aminoacyl-tRNA-synthetases (AlaRS, AsnRS, IleRS, PheRS1+2), *E. coli* Methionyl-tRNA formyltransferase (MTF), the translation elongation factor EF-Ts (which co-purifies with the MS2rep subunit), and the two translation initiation factors IF1 and IF3[29]. To identify TFs responsible for this marked effect on replicase activity, we performed selective depletion experiments with all LD2 proteins (Fig. 3a). Omitting the added EF-Ts led to a complete loss of activity, revealing that the excess amount of EF-Ts in the LD2 fraction is essential for transcription from the artificial [F30-Bro(−)]$_{UTR(−)}$ template. We further observed an unexpected influence on transcription activity by the initiation factors IF1 and IF3. While omission of IF1 caused a ~50% reduction in [F30-Bro(+)]$_{UTR(+)}$ synthesis, the absence of IF3 led to a >400% increase in transcription activity compared to the positive control

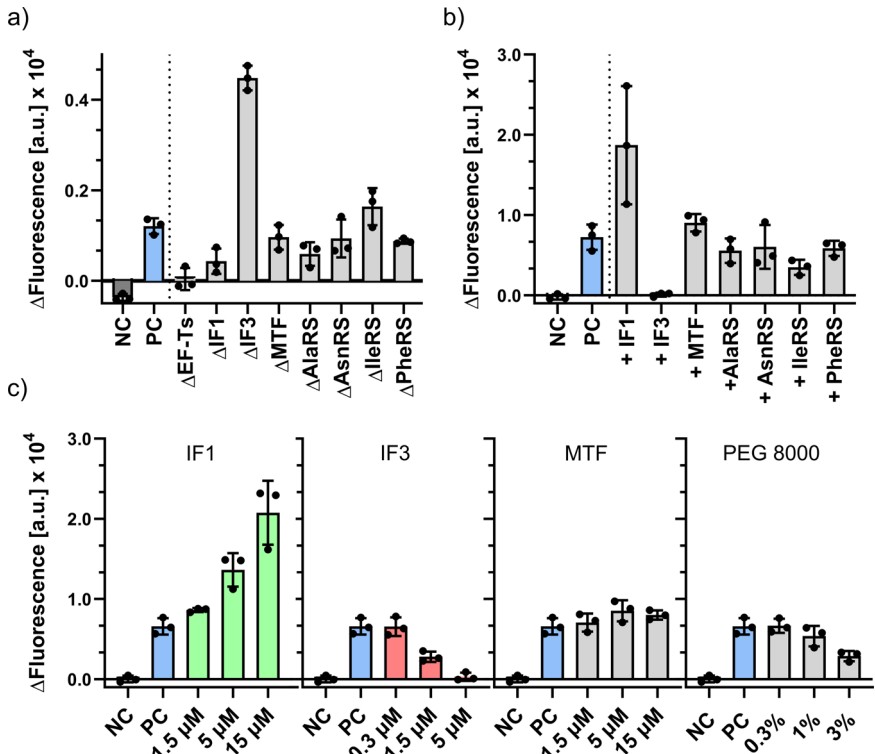

**Fig. 3 [F3O-Bro(+)]ᵤₜᵣ₍₊₎ transcription by MS2 replicase in the absence and presence of various PURE enzymes. a** DFHBI-1T fluorescence increase recorded after sample incubation at 37 °C for 1 h. Reactions contained 0.3 μM MS2rep, 50 nM [F3O-Bro(−)]ᵤₜᵣ₍₋₎, EF-Tu (15 μM), S1 (1.5 μM) and full LD2 (5 μM) except for the proteins indicated. Negative control (NC, dark grey) contained no MS2rep while the positive control (PC, blue) contained the full set of LD2 proteins and 0.3 μM MS2rep. **b** DFHBI-1T fluorescence as in **a** but in the presence of MS2rep and individual LD2 proteins alone (5 μM). The positive control (PC, blue) contained 0.3 μM MS2rep, 50 nM [F3O-Bro(−)]ᵤₜᵣ₍₋₎, EF-Tu (15 μM), EF-Ts (15 μM) and S1 (1.5 μM), and the negative control (NC) was identical with the exception of no MS2rep. **c** Co-factor titration assay. Reactions were carried out as in **b** but with varying concentrations of selected PURE proteins, or PEG 8000 to analyse excluded volume effects. Replication mixtures were programmed with 50 nM [F3O-Bro(−)]ᵤₜᵣ₍₋₎. Fluorescence was measured every minute over a 60-minute time course at 37 °C. Error bars indicate standard deviation based on three independent replicates prepared using the same protein stock solutions. The negative values for NC in panel **a** resulted from a baseline drift during the experiment.

containing the full 1x LD2 protein fraction. We excluded the possibility that the increase in fluorescence was a result from a direct interaction of the initiation factors with the aptamer domain of [F3O-Bro(+)]ᵤₜᵣ₍₊₎ since neither protein affected the fluorescence intensity of pre-synthesised [F3O-Bro(+)]ᵤₜᵣ₍₊₎ in separate control experiments (Supplementary Fig. 2a). Taken together, these findings indicate that IF1 stimulates MS2 replicase activity, whereas IF3 acts as an inhibitor. This hypothesis was further corroborated in additional experiments, in which an excess of 5 μM of each of the eight individually purified LD2 proteins was added to the reaction mixture (Fig. 3b). Whereas excess IF1 led to an increased transcription activity, adding an excess of IF3 completely abolished [F3O-Bro(+)]ᵤₜᵣ₍₊₎ synthesis. In contrast, the omission or supplementation of the four tRNA synthetases had no impact on MS2 replicase activity under these conditions. Both the inhibitory effect of IF3 and stimulating effect of IF1 showed clear dose dependence with observable effects even at low micromolar concentrations (Fig. 3c), supporting the notion that they are based on a direct functional interaction with the MS2 replicase core complex or the RNA template. In contrast, control experiments using MTF, PEG8000 or AlaRS at increasing concentrations showed that neither non-specific protein-protein interactions nor excluded volume effects are responsible for the observed modulation of replicase activity induced by IF1 and IF3 (Fig. 3c and Supplementary Fig. 2b).

**IF3 competes with MS2rep for RNA templates**. In further experiments, we found evidence that inhibition by IF3 is based on a

direct competition with the replicase for RNA binding as only an excess of the MS2rep complex could rescue transcriptional activity (Supplementary Fig. 3a). Indeed, previous studies reported that there is a sequence-specific interaction between IF3 and the 3′ end of MS2 RNA[33–35]. To further corroborate the hypothesis that IF3 acts as a competitive inhibitor, we measured the initial reaction velocities of [F3O-Bro(+)]ᵤₜᵣ₍₊₎ synthesis as a function of [F3O-Bro(−)]ᵤₜᵣ₍₋₎ input concentration in the presence of IF3. As expected, the addition of micomolar amounts of IF3 raised the apparent half-saturation concentration of MS2rep for the RNA template while not affecting the maximum rate of transcription at saturating [F3O-Bro(−)]ᵤₜᵣ₍₋₎ concentrations (Supplementary Fig. 3b). The apparent decrease of MS2rep for the template in presence of IF3 suggests that IF3 directly competes with MS2rep for the free 3′ end of the RNA template. Using fluorescence anisotropy of DHFB-1T bound to the template [F3O-Bro(+)]ᵤₜᵣ₍₋₎, which encodes the sense strand of the broccoli aptamer, and additional electrophoretic mobility shift assays (EMSAs), we determined an apparent dissociation constant $K_D$ of ~3.5 μM for the interaction between IF3 and [F3O-Bro(+/−)]ᵤₜᵣ₍₋₎. This affinity correlates well with the dose dependence observed for transcription inhibition of MS2rep by IF3 (Supplementary Fig. 4a, b).

**IF1 stimulates synthesis of the full-length MS2 genome**. IF1 can act as a general RNA chaperone that binds with moderate micromolar affinity to RNA[36–38]. We observed no increase in DFHBI-1T fluorescence anisotropy when IF1 was titrated to a [F3O-Bro(+)]ᵤₜᵣ₍₋₎, which is not surprising given the low

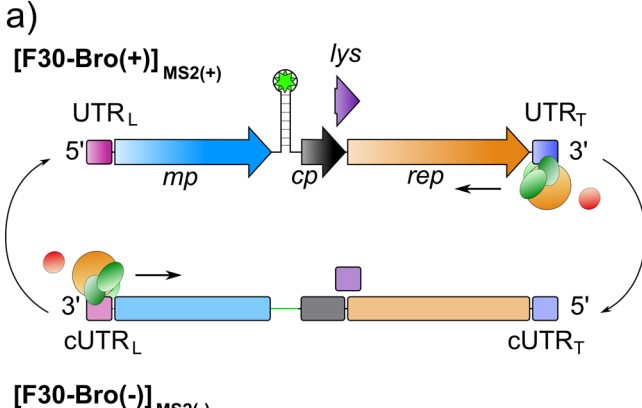

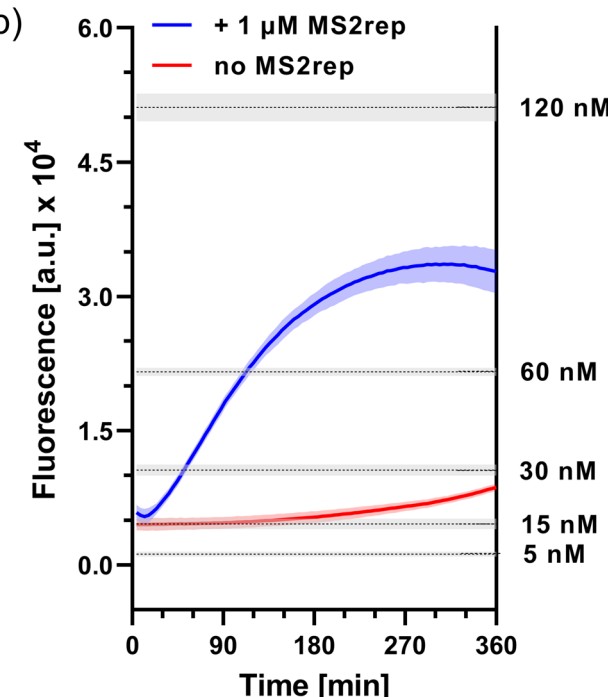

**Fig. 4 Full-length MS2 genome replication in PURExpress. a** Replication scheme of [F30-Bro(+/−)]$_{MS2(+/−)}$ RNA by the MS2rep complex. **b** Fluorescence change over the 6-h incubation at 37 °C for replication analysis of 15 nM [F30-Bro(+)]$_{MS2(+)}$ in PURExpress with (blue) and without (red) 1 μM MS2rep. Grey lines indicate average fluorescence levels of reference concentrations of in vitro transcribed [F30-Bro(+)]$_{MS2(+)}$ in PURExpress. Standard deviations are derived from three independent replicates.

molecular weight of IF1 (9.1 kDa) in relation to the RNA template (132 kDa). In EMSA experiments based on a Tris/acetate buffer system, the RNA bands showed only a mild smearing behaviour at increasing IF1 concentrations (Supplementary Fig. 4c). However, when the buffering system was changed from Tris/acetate to Tris/borate, a shift indicative of a direct RNA-protein interaction became also apparent for IF1 (Supplementary Fig. 4d). Together, these findings suggest that the enhancing effect of IF1 and the inhibiting effect of IF3 might rather be due to a direct protein-RNA interaction, and less likely due to an interaction with the replicase holo complex.

While [F30-Bro(+)]$_{UTR(+)}$ synthesis enables monitoring (+) strand synthesis from an artificial template, it provides no information on the complete replication cycle of the natural

~3600 nt MS2 genome. To probe full-length genome replication by the in vitro reconstituted MS2 replicase complex, we integrated the broccoli aptamer into the (+) strand of MS2 wild-type (MS2wt) genome at an amenable site downstream of the open reading frame of the maturation protein (Fig. 4a), where its insertion only minimally interferes with replication[39–41]. Using this construct [F30-Bro(+)]$_{MS2(+)}$, we observed a continuous increase in DFHBI-1T fluorescence when the MS2rep·-S1·EF-Ts complex was incubated in the PURExpress system, which suggests processive genome replication (Fig. 4b). Comparison with reference inputs of [F30-Bro(+)]$_{MS2(+)}$ showed an estimated sixfold amplification, corresponding to an increase of [F30-Bro(+)]$_{MS2(+)}$ from 15 nM to ~90 nM over 4 h.

Having shown that the MS2 replicase complex can replicate genomic MS2 RNA in the PURExpress system, we further sought to dissect the influence of the individual co-factors on the ability of the replicase to synthesise the genomic (+) and (−) strands (Fig. 5a, b). Synthesis of genomic (+) and (−) strands from the corresponding [F30-Bro(−)]$_{MS2(+/−)}$ templates produced only a weak fluorescence output compared to synthesis of [F30-Bro(+)]$_{UTR(+)}$ using [F30-Bro(−)]$_{UTR(−)}$ as template (Fig. 5c, d). Notably, the omission of an excess of S1 in these experiments did not significantly affect genome synthesis unlike for the shorter unnatural [F30-Bro(−)]$_{UTR(−)}$ construct used during initial experiments (Fig. 2a). This finding indicates that the bound S1 present in the purified complex is sufficient for effective replication of the natural replicase substrate.

The fluorescence output of the broccoli aptamer domain during genomic (−) strand synthesis was not affected by supplementation of IF1 (Fig. 5e). However, an in-gel fluorescence analysis revealed a drastic increase in RNA synthesis in the presence of the co-factor, with the majority of product migrating at the expected size of a full-length duplex (Supplementary Fig. 5). This suggests that newly synthesised single-stranded (−) strand RNA was scarce. In contrast, a stimulation of both overall RNA synthesis as well as broccoli fluorescence was observed during synthesis of [F30-Bro(+)]$_{MS2(+)}$ from [F30-Bro(−)]$_{MS2(−)}$ template in the presence of IF1 (Supplementary Fig. 5 and Fig. 5f). Thus, IF1 stimulated MS2 replicase activity independent from the polarity of the template.

In conclusion, the isolated MS2 replicase holocomplex was able to synthesise genomic RNA from both (+) and (−) strand templates in batch reactions. However, under these conditions, both strands readily anneal to form double-stranded RNA, which can no longer be used as a template for replication[42,43]. In contrast, replication of the full-length genome in PURE systems enabled sustained synthesis of the genomic (+) strand. Under these more in vivo-like conditions, ribosome binding and translation to the MS2(+) RNA may counteract direct annealing with newly synthesised (−) strand, thereby supporting continuous RNA replication similar to what has been described for in vitro replication by the Qβ replicase complex[44].

**Spontaneous formation of replicable RNA species.** The purified replicase complex of phage Qβ is well known for its spontaneous in vitro synthesis of rapidly amplifying RNA species of different length and nucleotide sequence, even in the absence of externally added template molecules[45–47]. To test if MS2rep is capable of a similar spontaneous generation of short amplifiable RNA species, we compared the activity of both purified Qβ and reconstituted MS2 core complex in template-free reactions supplemented with NTPs and SYBR Green nucleic acid stain. As expected, we observed a rapid increase in fluorescence after a brief lag phase of ~5–10 min in the presence of the Qβ heterotetramer (Supplementary Fig. 6a), suggestive of the rapid formation of small

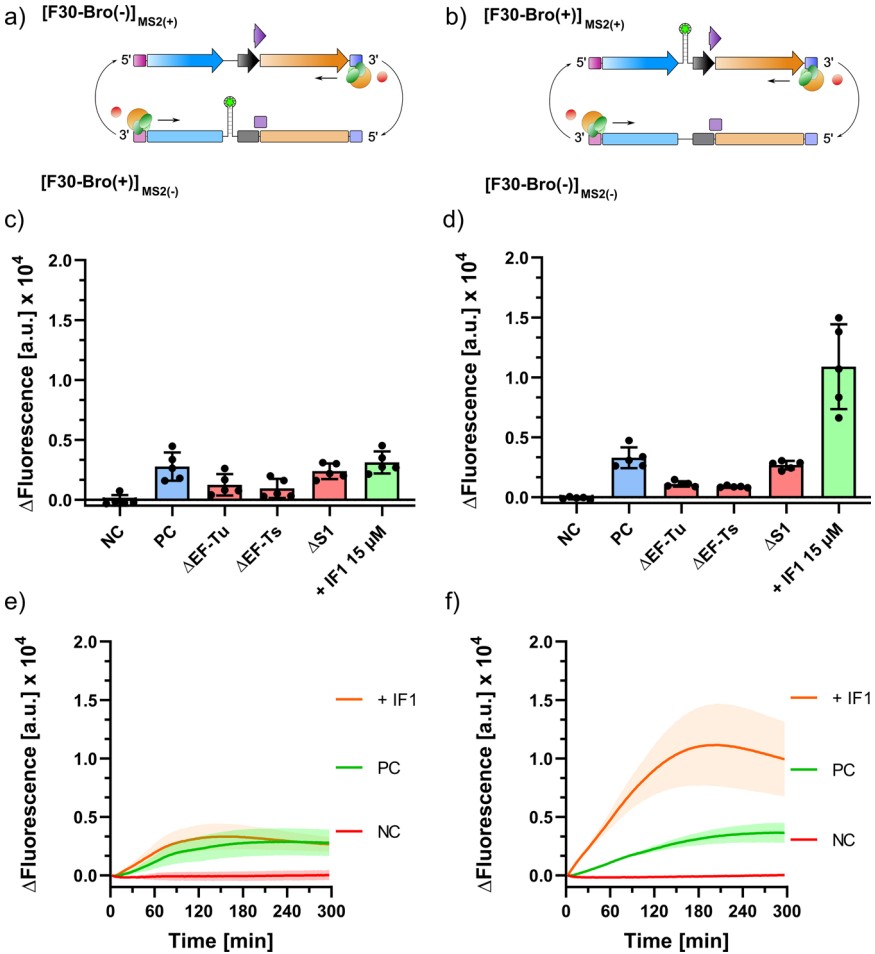

**Fig. 5 Replication of genomic (−) and (+) strand under isolated reaction conditions.** Replication and fluorescence readout scheme for **a** [F30-Bro(+)]$_{MS2(−)}$ and **b** [F30-Bro(+)]$_{MS2(+)}$ synthesis. DFHBI-1T fluorescence of **c** [F30-Bro(+)]$_{MS2(−)}$ and **d** [F30-Bro(+)]$_{MS2(+)}$ synthesis after 3 h incubation in reaction buffer supplemented with MS2rep (1 μM) in the absence (red) of essential co-factors EF-Tu and EF-Ts (15 μM each), S1 (1.5 μM), or in the presence of all co-factors and IF1 (15 μM) (green). Positive controls (PC, blue) contained all co-factors except IF1. Negative controls (NC, grey) contained no MS2rep. Initial non-fluorescent template concentrations were 50 nM. **e, f** show the fluorescence time traces of both reactions in **c, d** recorded over five hours. Negative control (NC, red), positive control (PC, green), additional 15 μM IF1 (+IF1, orange). Error bars indicate standard deviations calculated from independent replicates from the same protein stocks.

amplifying RNA species ("RNA parasites") described in previous studies[47]. We verified the efficient formation of small replicable RNAs by the Qβ replicase by gel electrophoresis (Supplementary Fig. 6b). Surprisingly, we observed no such spontaneous formation of RNAs when the MS2 replicase complex in the presence of IF1 was incubated for 75 min under the same conditions (Supplementary Fig. 6a, b). As the MS2 enzyme did not show a similar strong background activity as the Qβ core complex, we asked whether the enzyme was able to produce short amplifying RNAs as by-products during MS2 genome replication. To test this hypothesis, we performed serial transfer experiments with the MS2 replicase core complex (MS2rep·S1·EF-Ts·EF-Tu) in the absence and presence of IF1, IF3 and MS2(+) RNA (Supplementary Fig. 6c, d). In reactions with the full-length genome, we observed a rapid degeneration of the ~3600 nt RNA molecule during the first two dilutions concomitant with the emergence of smaller RNA species with a broad size distribution and a dominant RNA band migrating at ~200 nt. The emergence of aberrant RNA products was strongly increased when the MS2 core complex was further supplemented with IF1, and reduced when supplemented with IF3. Intriguingly, even in all control reactions without the input of MS2(+) RNA, the small RNA species emerged after the first serial transfer (Supplementary Fig. 6c, d).

For sequence properties of the newly evolved RNA replicators, we reverse transcribed, sequenced and analysed the reaction products. Notably, we obtained only a single clonal sequence from these experiments (MSRP-22), which showed almost perfect homology with the first 118 nt of the 5′ UTR and the last 105 nt of the 3′ UTR of MS2wt (Fig. 6a, b and Supplementary Table 3). We confirmed that MSRP-22 is a genuine RNA template for MS2 replicase since it was specifically amplified in an input concentration-dependent manner in batch reactions (Fig. 6c). In contrast, MS2rep was not able to amplify RQ135, a typical RNA parasite of Qβrep[48] within the same time window. This finding confirmed the differences in template requirements for the two phage replicases.

## Discussion

Previous attempts to characterise emesviral replicases in vitro were hampered by the general lability of the enzymes and the dependency of replicase activity on an unknown, yet easily dissociable, host factor[25,26]. Using the recombinant heterotrimeric MS2rep·S1·EF-Ts complex as a starting point, we elucidated the enigmatic properties of this archetypical class of RNA polymerase enzymes. Compared to the stable Qβrep·EF-Ts·EF-Tu·S1

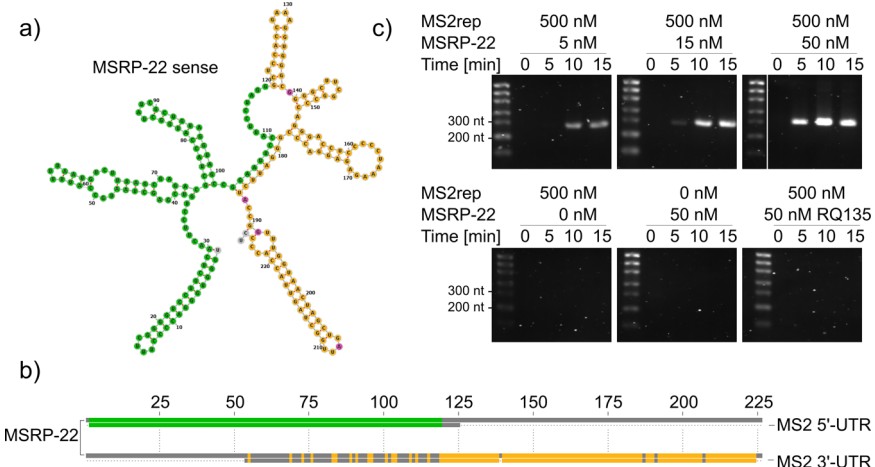

**Fig. 6 Properties of a selected small replicating RNA species. a** Minimum Free Energy structure of MSRP-22 predicted by RNAfold (default parameters)[60]. Colours indicate sequences aligning to the nucleotides 1–118 within the 5′ UTR (green) and nucleotides 3464–3568 within the 3' UTR (yellow) of MS2wt, respectively. Nucleotides annotated in pink indicate mismatches and grey nucleotides could not be aligned to MS2wt[60,61]. Annotation as sense or antisense strand follows the orientation of the matching MS2 genomic or antigenomic strand, respectively. **b** Sequence alignments of the isolated MS2 RNA parasite MSRP-22 with the 5′ UTR of MS2(+) (top, green) and 3′ UTR of MS2(+) (bottom, yellow), respectively. Alignments were made with Jalview (Version 2.11.1.3)[62]. **c** Time-course of replication of MSRP-22, analysed by gel electrophoresis on a 2% TAE agarose gel. Reaction mixtures contained varying concentrations of input MSRP-22 antisense RNA (5, 15 or 50 nM, image cropped) with 500 nM MS2rep core complex (top row). Controls contained either no input RNA, no MS2rep or MS2rep with RQ135 RNA as non-specific RNA input (bottom row). The ladder used was RiboRuler Low Range RNA Ladder (Thermo Fisher Scientific).

heterotetramer, which is capable of replicating a number of non-genomic RNAs as well as genomic Qβ(+) RNA in vitro[49], the MS2 replicase complex shows a number of different characteristics.

Firstly, EF-Tu, which is an integral part of the active complex for both replicases, only weakly binds to the stable MS2rep·S1·EF-Ts core trimer. Secondly, we identified IF1 as the previously unknown host factor improving MS2rep activity in vitro.

IF1 is a S1 domain protein that contains an oligomer binding (OB) fold found in a variety of RNA chaperones[37]. Indeed, IF1 acts as a transcription anti-termination factor, which can destabilise strong secondary structure elements and thereby facilitating RNA polymerase readthrough[38]. Furthermore, IF1 showed RNA chaperone activity during trans-splicing assays both in vivo and in vitro[36]. These properties and our own findings suggest that IF1 supports MS2 genome replication by destabilising secondary structure elements of single-stranded templates and exposing terminal and internal binding sites for the MS2 replicase and/or through facilitating replication initiation and product release. The influence of IF1 on RNA synthesis resembles the role of the OB proteins S1 and Hfq for (−) strand synthesis during the Qβ replication cycle. Here, Hfq facilitates the access of replicase to the 3′ end of the genomic (+) strand[50,51], while S1 appears to contribute to both termination of replication and re-initiation after product release[52]. Thus, a similar yet MS2-specific role of IF1 for the remodelling of MS2 templates seems plausible.

The strong inhibitory effect of IF3 on MS2 replicase activity seems counterintuitive at first and rather indicative of an in vitro artefact. However, early studies reported a specific interaction of IF3 with the 3′ terminus of MS2[33–35], which we reproduced and quantified in our study. Furthermore, our results suggest a direct competition for the 3′ end between the replicase and IF3 because increasing concentrations of either the replicase or the substrate RNA compensated for the inhibitory effect of IF3. Notably, MS2 gene expression was also shown to be completely independent from IF3, unlike for E. coli host proteins[53]. Thus, replication and translation of MS2 genomes seem to be well adapted to conditions under which IF3 levels are minimal, such as it was reported

for E. coli cells that have reached stationary phase[54]. While phage virions are still produced under these conditions, protein synthesis rates are usually insufficient to trigger cell lysis and phage release[55]. Therefore, we speculate that the ability of the MS2 replication system to continuously produce infectious but host-contained virions in slowly growing cells may enable phage persistence within starved bacterial populations. This state may last until growth conditions and therefore a lytic reproduction cycle can be restored. However, in initial experiments under laboratory growth conditions, we did not observe any noteworthy effect on MS2 infectivity upon overexpression of the initiation factors in a laboratory E. coli F+ strain. Neither the number nor the morphology of plaques were largely altered upon over-expression of IF1, IF3, and, as a control AsnRS (Supplementary Fig. 2c and Supplementary Data 2). Thus, a direct connection between the inhibitory effect of IF3 and the stimulating effect of IF1 on MS2 infectivity was not found under these simple laboratory conditions, either because both proteins were already present at saturating levels or because their effect is purely restricted to in vitro environments. Unfortunately, both initiation factors cannot be deleted from the E. coli genomes as they are absolutely essential for survival. Future studies aiming to investigate a potential role of IF3 or IF1 on MS2 infection might therefore require specific E. coli strains with e.g. heat-labile initiation factors[56]. Inhibition of genome replication by IF3 might also be part of well-orchestrated infection cycle of MS2. For example, the protein could serve as a simple negative feedback regulator for RNA replication when sufficient amounts of MS2 genome have been synthesised for translation and packaging.

The related Qβ replicase is notorious for the spontaneous generation of short, exponentially amplifying "RNA parasites" from trace amounts of contaminating RNA even in single batch reactions[45,47]. In contrast, the reconstituted MS2 replicase core complex required additional serial transfers before replicative RNA species were enriched in substantial amounts. While the emerging RNA population was dominated by a small sequence migrating at ~300 nt in agarose gels, it retained an overall broad size distribution even after five serial transfers. The lack of

convergence to a discrete number of small, dominant replicators (such as for Qβ replicase, Supplementary Fig. 6) implies either that initiation, rather than elongation, is critical for sequence replication, or that longer sequences are generated by mechanisms such as template switching or non-templated terminal transferase activity[57].

The reason for the generally lower tendency of MS2 replicase to rapidly generate and amplify non-genomic RNA species in comparison with Qβ replicase remains unclear. One explanation is an overall lower in vitro activity and/or lifetime of the holoenzyme similar to pioneering reports that used enriched protein fractions for activity assays[58]. Alternatively, MS2 replicase might be less promiscuous than Qβ replicase towards non-genomic templates and therefore less prone to generate short parasitic sequences. In agreement with this conjecture, the clonal sequence derived from the serial transfer experiment consisted exclusively of major parts of the 5′ and 3′UTR of the wild-type genome without additional sequence elements, suggesting that these motifs form the core of replicable units. Thus, both UTR elements may enable the design of replicable mRNA species, which could be used for dynamic in vitro evolution studies[18,19,45,46] or the generation of self-amplifying mRNAs in vivo[59].

## Methods

**Preparation of RNAs and plasmids**. The preparation of all RNAs and the cloning of all plasmids are described in detail in the Supplementary Methods. Supplementary Table 1 contains all primers used for RNA preparation and cDNA synthesis. Supplementary Data 4 lists all DNA templates used for IVT. Supplementary Data 5 contains the sequences of all plasmids used in this study.

**Preparation of proteins**. The preparation of proteins followed an adapted version of the protocol described by Shepherd et al.[29]. In short, Top10 or BL21(DE3) E. coli cells, respectively, (Thermo Fisher Scientific) were transformed by electroporation with either pBAD33-based (Top10) or pLD1-3 plasmids (BL21(DE3)). Proteins were overexpressed at 16 °C overnight in Lysogeny Broth (LB Lennox) with 0.2% L-Arabinose for all pBAD33-based constructs or 1 mM IPTG for pLD1-3, respectively. Subsequently, proteins were purified over HisPur™ Ni-NTA resin (Thermo Fisher Scientific) using HEPES buffer (50 mM HEPES·KOH pH 7.5, 250 nM NH$_4$Cl, 10 mM MgCl$_2$, 5 mM DTT) and stored in HEPES/glycerol buffer (50 mM HEPES·KOH pH 7.5, 100 mM KCl, 10 mM MgCl$_2$, 7 mM DTT, 30% Glycerol). Preparations from pLD1, pLD2 and pLD3 were adjusted to ten-fold stocks with the final protein content per reaction equalling concentrations as described previously[29]. A more detailed protocol is included in the Supplementary Methods. Supplementary Data 1 contains all mass spectrometry data used for protein identification of co-purified proteins in the MS2rep subunit preparation.

**Real-time fluorescence measurements**. All readout constructs are based on designs by Weise et al.[20]. For experiments using solution A (containing amino acids, tRNA, energy factors) and B (containing ribosomes and enzymes) of the PURExpress® In Vitro Protein Synthesis Kit (NEB), standard reactions were supplemented with 10 μM DFHBI-1T ((Z)-4-(3,5-difluoro-4-hydroxybenzylidene)-2-methyl-1-(2,2,2-trifluoroethyl)-1H-imidazol-5(4H)-one)[28], MS2rep and [F30-Bro(−)]$_{UTR(−)}$ or [F30-Bro(+/−)]$_{MS2(+/−)}$, respectively, as described for the individual experiments:

*Transcription of [F30-Bro(+)]$_{UTR(+)}$ in PURExpress*. 300 nM MS2rep, 150 nM [F30-Bro(−)]$_{UTR(−)}$ and 10 μM DFHBI-1T

*Transcription of [F30-Bro(+)]$_{MS2(+)}$ in PURExpress*. 1 μM MS2rep, 15 nM [F30-Bro(−)]$_{MS2(−)}$ and 10 μM DFHBI-1T

All experiments based on the in-house PURE system (PURE 3.0/ PUREred) were assembled including 0.5 mM each ATP, GTP, CTP and UTP, 10 mM DTT, 15 μM EF-Tu, 1.5 μM S1 and all other components as follows:

*Transcription of [F30-Bro(+)]$_{UTR(+)}$ in PURE 3.0*. 300 nM MS2rep, 150 nM [F30-Bro(−)]$_{UTR(−)}$, 1x LD1, LD2 and LD3 and 10 μM DFHBI-1T

*LD2 depletion experiments*. 300 nM MS2rep, 50 nM [F30-Bro(−)]$_{UTR(−)}$, 5 μM LD2 components and 10 μM DFHBI-1T

*Single LD2 component*. 300 nM MS2rep, 50 nM [F30-Bro(−)]$_{UTR(−)}$, 15 μM EF-Ts, 5 μM LD2 components and 10 μM DFHBI-1T

*Co-factor titration*. 300 nM MS2rep, 50 nM [F30-Bro(−)]$_{UTR(−)}$, 15 μM EF-Ts and 10 μM DFHBI-1T

*[F30-Bro]$_{MS2}$ replication*. 1 μM nM MS2rep, 50 nM [F30-Bro(−)]$_{MS2(+/−)}$, 15 μM EF-Ts, 10 mM DFHBI-1T and 0.5 U per μL RNase inhibitor (moloX)

The final concentration of MgCl$_2$ was 6 mM, with HEPES, KCl, and glycerol supplemented in 50 mM, 100 mM, and 18%, respectively. All reactions were prepared in MicroAmp Fast 8-TubeStrips (Thermo Fisher Scientific) and incubated at 37 °C in a StepOne Real-Time PCR System (Thermo Fisher Scientific). Fluorescence signals were recorded every 60 s over a total of four hours or every 120 s for six hours for experiments involving [F30-Bro(+/−)]$_{UTR(+/−)}$ or [F30-Bro(+/−)]$_{MS2(+/−)}$, respectively. Unless stated otherwise, all experiments were performed in independent triplicates assembled from the same stock solutions.

**Isolation and sequencing of MSRP-22**. To obtain sequencing data, RNAs from serial transfer experiments (see Supplementary Methods) were purified using Monarch® RNA Cleanup Kit (NEB) then polyadenylated using E. coli Poly(A) Polymerase (NEB) and re-purified with the same clean-up kit. Double-stranded cDNAs were synthesised using Template Switching RT Enzyme Mix (NEB) in combination with primers CDSII-T$_{24}$VN and TSO-CDSII (Supplementary Table 1) following the manufacturer's 2nd Strand cDNA Synthesis protocol. DNAs were purified using Monarch® PCR & DNA Cleanup Kit (NEB) and subsequently prepared for cloning with NEB® PCR Cloning Kit (NEB). For transformation, 2 μL of the reactions were transformed into chemically competent Top10 E. coli cells (Thermo Fisher Scientific).

The replication capability of MSRP-22 was analysed by gel electrophoresis. The samples for the replication time course were obtained by preparing the following reaction mixtures and incubating at 37 °C: 15 μM EF-Tu, 15 μM EF-Ts, 1.5 μM S1, 0.5 mM of each of ATP, GTP, CTP and UTP, and 10 mM DTT. The final concentration of MgCl$_2$ was 6 mM, with HEPES, KCl, and glycerol supplemented at 50 mM, 100 mM, and 18%, respectively. MS2rep was supplemented at 500 nM and RNAs in the concentrations described in Supplementary Fig. 6c, d.

**Preparation of gel electrophoresis samples**. All samples for gel electrophoresis were prepared by mixing equal volumes of reaction mixture and 2x RNA Gel loading dye (Thermo Fisher Scientific). Samples were heated to 70 °C for 5 min and then slowly cooled to room temperature. This allowed bound proteins to denature and dissociate from RNAs, as well as the complementary RNA strands to denature and anneal.

**Statistics and reproducibility**. The methods for statistical analysis and sizes of the samples (defined as n) are specified in both the results section and individual figure legends for all of the quantitative data. All values are presented as mean ± SD with the indicated sample size where applicable.

**Reporting summary**. Further information on research design is available in the Nature Research Reporting Summary linked to this article.

## Data availability

Supplementary Data 1 contains all mass spectrometry data used for protein identification. Source data for all graphs and charts prepared and analysed as part of this study can be found in Supplementary Data 2. Supplementary Data 3 contains the sequencing chromatogram for MSRP-22 cDNA. Further information is available from the corresponding author upon reasonable request.

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

## Acknowledgements

We thank K. Libicher for help with preparing the LD protein fractions. We thank the MPIB mass spectrometry core facility for the help with protein identification and analysis. We thank E. Y. Song for critically proofreading the manuscript. H.M. is grateful for funding by the European Research Council (ERC starting grant, RiboLife) under 802000.

## Author contributions

A.W. and H.M. wrote the paper; H.M. and A.W. designed the experiments and analysed and interpreted the data. L.I.W. contributed to the full-length genome replication experiments. All authors read and approved the final paper.

## Funding

## Competing interests

The authors declare no competing interests.
