## [Peer Review File · Communications Biology]

Reviewers' comments:

Reviewer #1 (Remarks to the Author):

Comments:

The manuscript entitled, "In vitro characterization.....replicase activity", by Wagner et al describes the identification of new host factors that modulate the RNA phage MS2 replicase functions in in vitro set-up using synthetic as well as genomic templates.

The most important take-away of this manuscript is to show the involvement of translation initiation factors, IF1 and IF3, in this replication process under in vitro conditions. However, the major limitations of this study are not providing experimental evidence to establish the mechanism of action of how IF1 and IF3 function in this process and the in vivo significance of this novel finding. Also, the manuscript at many places is too technical, which could be difficult for readers from outside the field to follow. I think the manuscript needs thorough revision in writing and additional experimentations to shed light on the mechanistic aspects as well as to understand the in vivo functional significance of the in vitro findings.

My specific comments are given below.

1. Line 96: Describe in detail the PURExpress kit.
2. Lines 163-164: How the indicated bands in figure S1 annotated? By mass-spec?
3. Line 167: How much sequence/structural similarities are there between Q β and MS2 replicases?
4. Lines 168-180: Need a more detailed description of the assays. What is DFHB1-IT? What are the differences between homemade and commercially available PURE kits? What are the components of these kits?
5. A positive control using only synthetic F30Bro (+) template is needed, where the fluorogen will readily bind even in the absence of the replicase function.
6. Line 185: Please describe the components of LD1, 2, 3. How they were prepared?
7. Please indicate PUREd in the figure 2 legend.
8. Line 202: Why fluorescence enhancement due to the depletions of LD1 and LD3 were not described?
9. It was not clear to me how each of the components was deleted in the replication assays.
10. Show direct binding of IF1 and IF3 to the replicase complex. Signal intensities of the gel bands of these two factors are very weak in figure S21.
11. Mechanism of inhibition by IF3 or stimulation by IF1 should be experimentally explored. Mere speculations in the discussion sections are not enough. Show the RNA binding abilities of these factors experimentally. Direct completion assays have to be performed to infer competitive inhibition by IF3.
12. It is very important to perform in vivo experiments with mutations in IF1 and IF3 to establish their functional involvement in the replication process.
13. The last section of the result appeared to be redundant for this manuscript. This manuscript should only focus on the roles of IF1 and IF3 in MS2 replicase functions.

Reviewer #2 (Remarks to the Author):

The manuscript "In vitro characterisation of the MS2 RNA polymerase complex reveals novel host factors that modulate leviviral replicase activity" by Wagner et al describes activities of two previously unknown host factors IF1 and IF3 of phage MS2 replicase complex.

In general, the article is well written and could be suitable for publication in Communications Biology, but there are a few issues, which should be addressed as listed below.

From Fig 3A it is evident that omission of EF-Ts led to near-complete loss of activity. However, EF-Ts was co-purified with replicase, therefore it is not clear, why enzyme was completely inactive.

In the same Figure 3A it seems that depletion of IF1 in fact had essentially the same effect as depletion of AlaRS – within the error margin. Although from Fig 3B it does not seem that addition of AlaRS has any effect, perhaps a titration experiment, shown in Fig 3C should be done also for AlaRS.

In Figure B another control – namely MS2 core complex without template could be included. Also, could be worth including similar experiments with IF3.

From Supplementary Figure 1 it seems that Hfq was co-purified with MS2 replicase, but not Qbeta replicase. I find this rather intriguing, since it has been shown earlier that Hfq enhances replication of Qbeta. Authors should comment on this – maybe after all Hfq is to some extent beneficial also for MS2 replicase.

In reference #5, Stock-Ley, P.G should be Stockley, P.G.

In reference #37, author names are wrong.

Reviewer #3 (Remarks to the Author):

The single-stranded RNA bacteriophages (ssRNA phage) MS2 and Qbeta have been model systems for modern molecular biology. While MS2 is the first organism to have its genome sequenced, Qbeta has been extensively studied for RNA-dependent RNA replication and in vitro evolution. One reason is because it appears the Qbeta replicase is more stable and easier to work with than the MS2 replicase. In Qbeta, it has been estimated that the Qbeta genome encodes a beta-subunit of the replicase, along with the host proteins, EF-Tu EF-Ts and the ribosomal protein S1, from a holoenzyme and another host protein hfq is needed for the proper replication of the phage genomic RNA. Since then it has been assumed the paradigm for the ssRNA phage genome replication. However, whether there is another host factor for MS2, replacing the hfq for Qbeta, was not known. In this manuscript by Wagner et. al., the authors seek to identify the missing host proteins required for the assembly of the active replicase complex for MS2. The authors found that the translation initiation factors IF1 and IF3 directly modulate MS2 replicase activity. While IF1 enhances the genomic RNA replication, IF3 inhibits the RNA replication. In addition, the authors determined the minimal RNA sequence motifs required for MS2 replicase-dependent amplification. These results can be impactful and may provide a basis for a new evolution platform based on MS2 replicase. I have the following comments and issues for the authors to address before I can fully endorse its publication.

Major issues:

The authors use an assay, in which a fluorescence readout is produced by (+) strand synthesis of the broccoli aptamer from a (-) strand template that is fused with the 3'-end of genomic MS2 (-) strand (F30-Bro(-)), in the presence of the fluorogen DFHBI-1T. They then observed an enhancement of the fluorescence signal when adding IF1 to the MS2 replicase and a suppression of the fluorescence after adding IF3. However, there is not a control experiment to see if adding IF1 alone, without the MS2 replicase complex, would already increase the fluorescence of the aptamer and vice versa for IF3. Since IF1 and IF3 interact with the RNA, is it possible that it is simply their interaction with the aptamer RNA that modulate the fluorescence signal? Therefore, a control experiment is needed.

Minor issues:

In Figure 1, the cartoon scheme shows the unknown factor bind to the replicase holoenzyme. This is misleading, for even in Qbeta, it has not been demonstrated that the host factor hfq has direct interaction with the Qbeta replicase.

In the title: "In vitro characterization of the MS2 RNA polymerase complex reveals novel host factors that modulate leviviral replicase activity", the word "novel" is deceiving. Since both IF1 and IF3 are not novel proteins inside the cell. When I read this title, I was expecting some new proteins to be identified to work with the MS2 replicase, but a bit disappointed when the factors are IF1 and IF3.

Reviewer #1

The manuscript entitled, "In vitro characterization.....replicase activity", by Wagner et al describes the identification of new host factors that modulate the RNA phage MS2 replicase functions in in vitro set-up using synthetic as well as genomic templates. The most important take-away of this manuscript is to show the involvement of translation initiation factors, IF1 and IF3, in this replication process under in vitro conditions. However, the major limitations of this study are not providing experimental evidence to establish the mechanism of action of how IF1 and IF3 function in this process and the in vivo significance of this novel finding. Also, the manuscript at many places is too technical, which could be difficult for readers from outside the field to follow. I think the manuscript needs thorough revision in writing and additional experimentations to shed light on the mechanistic aspects as well as to understand the in vivo functional significance of the in vitro findings.

My specific comments are given below.

1. Line 96: Describe in detail the PURExpress kit.

The commercially available PURExpress system consists of two solutions: Solution A contains the metabolic and Solution B the enzymatic components required for coupled in vitro translation/transcription with an energy recovery system. The exact composition of the kit, which goes beyond the original protocol for the PURE system published in 2001, has not been disclosed by the supplier. Therefore, we cannot describe the kit in detail. We have now included a rough description of the PURExpress system in the Materials and Methods section.

2. Lines 163-164: How the indicated bands in figure S1 annotated? By mass-spec?

Yes. These bands were annotated by mass spectrometry. We have now included the MS raw data used for identification of the proteins in the Supplemental Data.

3. Line 167: How much sequence/structural similarities are there between Q β and MS2 replicases?

We have now added the result of a pairwise sequence alignment. The sentence now reads:

We also used a simplified enzyme mix containing only ribosomal protein S1 and EF-Tu because we did not expect ribosomes or kinase enzymes to be the missing co-factors based on the homology between MS2 and Q β (30). Both replicases amount to 30.5 % sequence identity and 43.2 % sequence similarity as determined by the EMBOSS Needle pairwise sequence alignment tool (31).

4. Lines 168-180: Need a more detailed description of the assays. What is DFHB1-IT? What are the differences between homemade and commercially available PURE kits? What are the components of these kits?

We thank the referee for pointing out this missing information. DFHB1-T stands for the compound (*Z*)-4-(3,5-difluoro-4-hydroxybenzylidene)-2-methyl-1-(2,2,2-trifluoroethyl)-1*H*-imidazol-5(4*H*)-one), which was developed as a superior fluorophore for spinach-based aptamers. We have now also included the appropriate reference to the manuscript by Song *et al.*, where the DFHB1-T was described first (<https://doi.org/10.1021/ja410819x>).

As described in our reply to 1.), we cannot provide the exact content of the commercially available PURExpress system. However, for a more detailed insight, we would like to kindly point the referee to the publication by Shimizu et al., which the PURExpress system is based on (<https://doi.org/10.1038/90802>). PURE 3.0 is the community-built version of the PURE system, which has a defined and known composition. We had already added a detailed overview of all components in the Supplementary Table 4.

5. A positive control using only synthetic F30Bro (+) template is needed, where the fluorogen will readily bind even in the absence of the replicase function.

We thank the referee for this suggestion. The fluorogenic role of already synthesised F30-Bro(+) (now referred to as $[F30\text{-Bro}(+)]_{\text{UTR}(+)}$) was already established in our previous publication by Weise *et al.* We have now also included this control into this manuscript along with the controls that demonstrate no altered fluorescence of pre-synthesised $[F30\text{-Bro}(+)]_{\text{UTR}(+)}$ in presence of IF1 and IF3. This additional data is now shown in Supplementary Figure 2a and verifies that the RNA is readily fluorescent in presence of the fluorogen and that neither IF1 nor IF3 do affect the fluorescence intensity. We also show in Figure 4b the same control for the full length $[F30\text{-Bro}(+)]_{\text{MS2}(+)}$ RNA in PURExpress (red line, 15 nM RNA).

6. Line 185: Please describe the components of LD1, 2, 3. How they were prepared?

A complete list of the individual components of LD1, LD2 and LD3 is shown in the Supplementary Information in Table 4. The purification scheme (standard protein purification of co-expressed proteins) is described in the method section in the main text, as well as in a more detailed version in the Supplementary Information under Supplementary Methods. We have also rewritten said paragraphs to now better explain the production and purification of LD1, LD2, and LD3.

7. Please indicate PUREred in the figure 2 legend.

We have amended Figure 2 and its legend accordingly.

8. Line 202: Why fluorescence enhancement due to the depletions of LD1 and LD3 were not described?

We now address this possible enhancement directly in the corresponding paragraph but would like to point out that the data shows a strong batch-to-batch variation in this respect (note the large error bars). It has been shown that an increase in molecular crowding can lead to a decrease in RNA-stability (<https://doi.org/10.1093/nar/gkz019>) and / or may alter the unwanted association between transcript and template. Presumably, the omission of LD1 or LD2 has a similar yet small effect, which according to our data may not even be significant.

We have now re-written the paragraph as follows (lines 183-190):

In addition to the expected dependency of the replicase on S1 and EF-Tu, we also observed an unforeseen impact on $[F30\text{-Bro}(+)]_{\text{UTR}(+)}$ synthesis upon omission of the individual LD protein fractions. Omission of both LD1 and LD3 had a modest enhancing effect on MS2rep activity which may not even be significant. This is likely caused by the altered crowding conditions in the absence of the protein factors, which may affect RNA folding and / or the formation of inert duplexes (32). In contrast, omission of LD2 caused a drastic loss of $[F30\text{-$

Bro(+)]UTR(+) transcription activity (Figure 2C) suggesting that at least one of its protein components enhances for MS2rep activity.

9. It was not clear to me how each of the components was deleted in the replication assays.

We apologise if this may have been unclear. To “delete” proteins from the individual LD-mixtures, they were simply omitted when the fractions were reconstituted from the different purified proteins. We have now reworded the corresponding paragraph in the text, to make this point clearer.

10. Show direct binding of IF1 and IF3 to the replicase complex. Signal intensities of the gel bands of these two factors are very weak in figure S21.

We assume the reviewer was referring to Supplementary Figure 1 (there was no figure S21), which shows the protein preparations post purification by affinity chromatography, and more specifically to the lane with purified MS2rep subunit. However, in this lane, neither IF1 nor IF3 were annotated, as we did only annotate bands that were a) visible as distinct bands with reasonable intensity and/or b) convincingly confirmed by mass spectrometry and peptide fingerprinting. Both of this was not the case for IF1 and IF3.

Instead, we now provide direct evidence that both factors act by binding to the template RNA rather than to the replicase complex similar as it has been described for Hfq in the case of Qbeta replicase. For both proteins we can confirm template binding by band shift assays in native gels. Furthermore, we have now additional data, which show that IF3 binds template RNA with micromolar affinity and acts as a competitive inhibitor for MS2 replicase (see next comment).

11. Mechanism of inhibition by IF3 or stimulation by IF1 should be experimentally explored. Mere speculations in the discussion sections are not enough. Show the RNA binding abilities of these factors experimentally. Direct completion assays have to be performed to infer competitive inhibition by IF3.

We thank the referee for this suggestion. We have now included additional experiments to back up our speculation in the discussion part. Using both band shift assays as well as fluorescence polarisation experiments, we can now show that IF3 does indeed directly bind to MS2, or MS2-derived RNA and shows characteristics of a competitive inhibitor for replicase initiation and / or elongation. This new data is now shown in Supplementary Figure 3 and well in agreement with previous studies, which report a direct and specific interaction of IF3 with 3'-termini of MS2-derived RNA.

IF1 seems also to weakly interact with RNA as shown by band shift assays (Supplementary Figure 4). The protein is too small for fluorescence polarisation experiments that are based on DFHB-1T fluorescence. The moderate affinity of IF1 for RNA is well in agreement with previous studies that report an RNA-chaperone like effect for IF1, which, amongst other things, facilitates RNA polymerase readthrough.

We have included these new findings in the revised version of the manuscript.

12. It is very important to perform *in vivo* experiments with mutations in IF1 and IF3 to establish their functional involvement in the replication process.

We agree that a further *in vivo* involvement of these two proteins during the MS2 life cycles is of great interest for microbiologists. However, in our opinion, the referee request goes beyond the scope of this study, which is aimed at reconstituting the active MS2 replicase *in vitro*, which had not been achieved since the 1960s despite several efforts.

One of the main reason of which addressing this microbiological question is difficult is that both IF1 and IF3 are absolutely essential housekeeping proteins for *E. coli*, which can not simply be mutated without massively causing pleiotropic effects such as growth defects or temperature sensitivity (compare for example <https://doi.org/10.1007/BF00446908> or <https://dx.doi.org/10.1128%2Fjb.176.1.198-205.1994>). Trying to create a viable IF1/3 variant, which is also still able to grow and enable complex infection studies with MS2 would requires excessive genetic engineering and variant screening. We have tried to overexpress both wild type initiation factors in *E. coli* under laboratory conditions to see if there is any effect on MS2 infectivity. However, in these very initial experiments we failed to see either a decrease or increase in MS2 infectivity (as measured by plaque forming units), presumably because both proteins are already highly expressed in exponentially growing *E. coli* cells in the first place. Nevertheless, we have included these experiments in the new Supplementary Figure 2c as well as in the Supporting Source Data and extended our discussion on that topic.

We recognise that the referee is highly interested in a more detailed *in vivo* study of the host factor / replicase interaction (so are we to some extend). However, we hope that the referee also understands that the focus of our lab is cell-free synthetic biology and the reconstitution of phage replication system rather than the biology of the host-phage interaction itself. Yet, we also believe that other people more interested in the subject might find our findings helpful and useful for their research.

13. The last section of the result appeared to be redundant for this manuscript. This manuscript should only focus on the roles of IF1 and IF3 in MS2 replicase functions.

We thank the referee for his/her opinion on this topic but respectfully disagree with the notion that this is redundant. The *de novo* synthesis of a short replicating RNA that is functional *in vitro* has never been described before for MS2. So far, such RNA parasites have only been described for Qbeta replicase. We have now also analysed the effect of IF1 and IF3 on the spontaneous generation of self-replicating RNAs such as MSRP-22, which now adds to the roles of IF1 and IF3 on MS2 replicase function. We also moved the previous Figure 6 to the SI to streamline the structure of the manuscript.

Reviewer #2

1. From Fig 3A it is evident that omission of EF-Ts led to near-complete loss of activity. However, EF-Ts was co-purified with replicase, therefore it is not clear, why enzyme was completely inactive.

We agree that this observation is somewhat unexpected. We assume that the amount of EF-Ts which is co-purified with both EF-Tu and the MS2rep subunit is below the level required for detectable enzymatic activity under dilute conditions as the affinity might be in the upper micromolar range. Thus, the MS2rep / EF-Ts complex might be too dynamic to enable transcription of non-natural templates under non-saturating EF-Ts concentrations (translation factors are amongst the highest concentrated proteins inside the bacterial cell). Alternatively, the copurifying EF-Ts / EF-Tu complex might be trapped in a non-functional conformation.

2. In the same Figure 3A it seems that depletion of IF1 in fact had essentially the same effect as depletion of AlaRS – within the error margin. Although from Fig 3B it does not seem that addition of AlaRS has any effect, perhaps a titration experiment, shown in Fig 3C should be done also for AlaRS.

We thank the referee for pointing this out. We have now included the suggested control experiment (new Supplementary Figure 2b). The new titration experiment confirms that AlaRS does not have any significant effect MS2rep activity.

3. In Figure B another control – namely MS2 core complex without template could be included. Also, could be worth including similar experiments with IF3.

We thank the referee for this suggestion. We have now added the additional controls (now included in Supplementary Figure 6c, d). In all cases we see formation of small, presumably continuously replicating RNAs similar as described for Qbeta: the MS2 replicase alone in absence of template and IF1 forms parasites only after extended incubation (3 h) and only at elevated replicase concentrations (1 μ M). In presence of IF3, this activity is severely repressed although we still observe formation of small RNAs that migrate at the same molecular weight as observed RNAs for conditions without IF3.

4. From Supplementary Figure 1 it seems that Hfq was co-purified with MS2 replicase, but not Qbeta replicase. I find this rather intriguing, since it has been shown earlier that Hfq enhances replication of Qbeta. Authors should comment on this – maybe after all Hfq is to some extent beneficial also for MS2 replicase.

We thank the reviewer for this interesting question. According to Su *et al.* (<https://doi.org/10.1006/viro.1996.8302>), the deletion of Hfq has no reducing effect on MS2 replication *in vivo*. However, we cannot exclude that there is a certain amount of redundancy in such that both IF1 and Hfq support MS2 replication with their RNA chaperone effect. The co-purification of Hfq might also simply reflect the higher amount of RNA-contamination in the MS2rep preparation, which would naturally lead to co-purification of a certain amount of

Hfq. Finally, we would also like to point out that the bands co-eluting with Qbeta replicase were not further analysed by MS, which is why we cannot rule out that the Hfq also co-purifies with this enzyme.

5. In reference #5, Stock-Ley, P.G should be Stockley, P.G.
In reference #37, author names are wrong.

We have corrected the references accordingly

Reviewer #3:

1. The authors use an assay, in which a fluorescence readout is produced by (+) strand synthesis of the broccoli aptamer from a (-) strand template that is fused with the 3'-end of genomic MS2 (-) strand (F30-Bro(-)), in the presence of the fluorogen DFHBI-1T. They then observed an enhancement of the fluorescence signal when adding IF1 to the MS2 replicase and a suppression of the fluorescence after adding IF3. However, there is not a control experiment to see if adding IF1 alone, without the MS2 replicase complex, would already increase the fluorescence of the aptamer and vice versa for IF3. Since IF1 and IF3 interact with the RNA, is it possible that it is simply their interaction with the aptamer RNA that modulate the fluorescence signal? Therefore, a control experiment is needed.

We thank the referee for this important suggestion. We have now included these controls into this manuscript, which demonstrate that there is no altered fluorescence of pre-synthesised [F30-Bro(+)]_{UTR(+)}, formerly described as F30-Bro(+) in presence of IF1 and IF3. This additional data is now shown in Supplementary Figure 2a. In addition, we have now included data from bandshift and fluorescence polarisation experiments that verify binding of IF1 and IF3 to [F30-Bro(+)]_{UTR(-)} (new Supplementary Figure 4). Thus, while both proteins bind to the RNA directly, they have no impact on the fluorescence of the aptamer-fluorogen complex.

2. In Figure 1, the cartoon scheme shows the unknown factor bind to the replicase holoenzyme. This is misleading, for even in Qbeta, it has not been demonstrated that the host factor hfq has direct interaction with the Qbeta replicase.

We thank the referee for pointing out this ambiguity. The “unknown” host factor was shifted in the scheme so that it floats next to the replicase and no longer implies it interacts directly with the holoenzyme.

3. In the title: “In vitro characterization of the MS2 RNA polymerase complex reveals novel host factors that modulate leviviral replicase activity”, the word “novel” is deceiving. Since both IF1 and IF3 are not novel proteins inside the cell. When I read this title, I was expecting some new proteins to be identified to work with the MS2 replicase, but a bit disappointed when the factors are IF1 and IF3.

We apologies for this overstatement. The “novel” was now removed from the title.

REVIEWERS' COMMENTS:

Reviewer #1 (Remarks to the Author):

The authors have addressed all the concerns that I raised in my review, and accordingly, the manuscript has been thoroughly revised. It is now significantly improved and should be accepted for publication.

Reviewer #3 (Remarks to the Author):

The authors have addressed all my concerns. I now support its publication.